# Are the Objectives Proposed by the WHO for Routine Measles Vaccination Coverage and Population Measles Immunity Sufficient to Achieve Measles Elimination from Europe?

**DOI:** 10.3390/vaccines8020218

**Published:** 2020-05-13

**Authors:** Pedro Plans-Rubió

**Affiliations:** 1Public Health Agency of Catalonia, Department of Health of Catalonia, Barcelona 08005, Spain; pedro.plans@gencat.cat; 2Ciber of Epidemiology and Public Health (CIBERESP), Madrid 28028, Spain

**Keywords:** measles, vaccination, immunity, measles vaccination coverage objectives, population measles immunity objectives, measles elimination, measles prevention, herd immunity

## Abstract

*Background:* The World Health Organization (WHO) proposed two-dose measles vaccination coverage of at least 95% of the population and percentages of measles immunity in the population of 85%−95% in order to achieve measles elimination in Europe. The objectives of this study were: (1) to determine the measles vaccination coverage required to establish herd immunity against measles viruses with basic reproduction numbers (Ro) ranging from 6 to 60, and (2) to assess whether the objectives proposed by the WHO are sufficient to establish herd immunity against measles viruses. *Methods:* The herd immunity effects of the recommended objectives were assessed by considering the prevalence of protected individuals required to establish herd immunity against measles viruses with Ro values ranging from 6 to 60. *Results:* The study found that percentages of two-dose measles vaccination coverage from 88% to 100% could establish herd immunity against measles viruses with Ro from 6 to 19, assuming 95% measles vaccination effectiveness. The study found that the objective of 95% for two-dose measles vaccination coverage proposed by the WHO would not be sufficient to establish herd immunity against measles viruses with Ro ≥ 10, assuming 95% measles vaccination effectiveness. By contrast, a 97% measles vaccination coverage objective was sufficient to establish herd immunity against measles viruses, with Ro values from 6 to 13. Measles immunity levels recommended in individuals aged 1−4 years (≥85%) and 5−9 years (≥90%) might not be sufficient to establish herd immunity against most measles viruses, while those recommended in individuals aged 10 or more years (≥95%) could be sufficient to establish herd immunity against measles viruses with Ro values from 6 to 20. *Conclusion:* To meet the goal of measles elimination in Europe, it is necessary to achieve percentages of two-dose measles vaccination coverage of at least 97%, and measles immunity levels in children aged 1−9 years of at least 95%.

## 1. Introduction

Measles is a highly contagious infectious disease associated with outbreaks, hospitalizations, and deaths in Europe, as well as in other regions of the world, despite highly effective vaccines being available. The main strategy to achieve measles elimination is based on high percentages of routine measles vaccination during childhood. In 2015, the European Region of the World Health Organization (WHO) renewed their commitment to the elimination of measles by the year 2020 [1]. In the WHO European region, the mean vaccination coverage for the first dose of the measles vaccine increased from 76.8% in 1980 to 92%−94% since 2003, and the mean vaccination coverage for the second dose of the measles vaccine increased from 68% in 1995 to 91%−92% since 2004 [2]. The mean vaccination coverage with two doses of the measles vaccine increased from 58.6% in 1995 to 84%−88% since 2004 [2]. Measles vaccines have decreased the incidence and mortality from measles in Europe, but measles cases and outbreaks are still occurring. 

The elimination of measles from Europe is feasible because humans are the only reservoir for measles, effective vaccines are available, highly sensitive, and specific diagnostic tests are available, and endemic measles transmission has been interrupted in America [3,4]. 

Unfortunately, measles cases and outbreaks increased from 2015 to 2019 [4,5,6,7,8]. In 2015, a total of 9010 measles cases, 4259 hospitalizations, and 2 deaths due to measles were reported by European countries to the WHO’s centralized information system for infectious diseases (CISID) [5,6,7]. In 2019, 16,485 measles cases, 9824 hospitalizations, and 8 deaths due to measles were reported by European countries to the CISID [5,6,7]. In 2018, 12,352 measles cases and 34 deaths due to measles were reported by countries of the European Union to the European Center for Disease Prevention and Control (ECDC) [8]. In 2019, 13,207 measles cases and 10 deaths due to measles were reported to the ECDD [9]. 

In the European Union in 2019, 28% of cases occurred among children under five years, 17% occurred among children 6–14 years old, and 55% occurred among individuals aged 15 or more years—71% of cases were unvaccinated and 18% had received one dose of the measles vaccine [9]. The overall incidence of measles was above the elimination target (one case per million population) in 29 (97%) countries of the European Union in 2019 [9]. 

Several factors may explain why measles continues to persist in Europe, including low measles vaccination coverage with two doses of the measles vaccine, low anti-measles immunity levels in areas and population groups (immunity gaps), mobility of individuals with measles across Europe, and loss of public confidence in vaccines [2,10,11]. Countries from the eastern part of Europe conducted supplementary vaccination activities to vaccinate population cohorts that were susceptible to measles, but they were not enough to achieve the vaccination coverage required to block measles transmission in Europe [11]. A recent study found that low percentages of two-dose measles vaccination coverage during 2015−2017 could be one of the factors explaining the persistence of measles during 2017−2018 in Europe [12]. 

The strategy plan proposed by the World Health Organization to achieve measles elimination in Europe by the year 2020 was based on four main measures [4,9,13]:Achieve and maintain routine measles vaccination coverage with two doses of the measles-mumps-rubella (MMR) vaccine ≥95%.Supplementary immunization activities to population groups at risk for measles and to individuals susceptible to measles.Intensive epidemiological surveillance.Rigorous outbreak control.

The objective of this strategy was to protect vaccinated individuals and to achieve and maintain a proportion (or prevalence) of protected individuals sufficient to establish the necessary herd immunity to block measles transmission in the community. For this reason, the strategic framework for the elimination of measles in the European Region proposed the following objectives for measles immunity in terms of proportion (or prevalence) of protected individuals: ≥85% in children aged 1–4 years, ≥90% in individuals aged 5–9 years, and ≥95% in individuals aged 10–14 years, 15–19 years, and ≥20 years [4] (Table 1). The objectives in terms of the prevalence of susceptible individuals were: ≥15% in children aged 1–4 years, ≥10% in individuals aged 5–9 years, and ≥5% in individuals aged 10−14 years, 15–19 years, and ≥20 years [14]. 

The age groups selected by the WHO correspond to preschool children (1–4 years), primary school children (5–9 years), secondary school children (10–14 years), and adults (≥15 years). Anti-measles immunity protection (or susceptibility) in different age groups can be assessed by means of developing serological surveys in representative samples of the population [15,16]. The WHO assumed that recommended anti-measles immunity levels were sufficient to establish the herd immunity required to block measles transmission in the community [14,17].

In Europe, routine measles vaccination is based on two doses of the MMR vaccine. Children receive their initial measles vaccine when they are 12–15 months old and the second dose when they are 3–15 years old [18]. Measles vaccination is based on two doses of the vaccine to provide protection to children that did not respond to their first dose (primary vaccination failure) and to reduce waning vaccine-induced immunity (secondary vaccination failure) [19].

The WHO and the ECDC recommend supplementary vaccination activities to close immunity gaps in adolescents and adults who have missed vaccination opportunities in the past [4,9]. Supplementary vaccination activities to vaccinate population cohorts that were susceptible to measles have been developed in countries from the eastern part of Europe [11], but screening and vaccination programs have not been implemented at either the national or regional level within Europe [20,21]. 

Sustained high vaccination coverage with two doses of the vaccine is the key preventive measure to successfully eliminating measles from Europe [4,9,21]. European countries have developed routine measles vaccination programs using one dose of vaccine since 1980, and routine measles vaccination using two doses of vaccine since 1995 [2]. Successful measles vaccination programs provide both direct protective effects among vaccinated individuals and indirect protective effects from herd immunity among unvaccinated individuals [22]. 

The establishment of herd immunity in the population depends on both the measles vaccination coverage and the following factors: (1) the effectiveness of the measles vaccination, (2) the duration of vaccine-induced immunity, (3) the average number of measles cases generated by a measles case in a totally susceptible population (basic reproductive number), and (4) measles immunity levels in unvaccinated population groups. The objectives of this study were: (1) To determine the measles vaccination coverage and measles seroprevalence required to establish herd immunity against measles viruses, and (2) to assess whether the objectives proposed by the WHO for routine measles vaccination coverage and population anti-measles immunity are sufficient to establish herd immunity against different measles viruses.

## 2. Materials and Methods 

### 2.1. Critical Measles Vaccination Coverage Associated with Herd Immunity against Measles Viruses

In this study, the herd immunity thresholds in terms of critical vaccination coverage (V_c_) and the critical prevalence of positive anti-measles serologic results (pc) were determined for different measles viruses and different values of measles vaccination effectiveness. Herd immunity against measles is defined as the indirect protection of susceptible individuals brought about by the presence of individuals with measles immunity in the population. The generation of measles epidemics depends on the average number of individuals directly infected (secondary cases) by one infectious case during the entire infectious period, when the infectious agent has entered a totally susceptible population [23]. This number is called the basic reproductive number (Ro), and epidemics occur when Ro is higher than 1. Anderson and May found values of Ro for measles viruses ranging from 12 to 18 in a review carried out in 1991 [23], but a recent review of studies assessing the Ro values found Ro values ranging from 6 to 45 in Europe, and from 6 to 60 in different countries of the world [24]. 

The chain of infection is blocked for infectious diseases transmitted person-to-person when the prevalence of protected individuals (I) is higher than a disease-specific critical proportion or prevalence of protected individuals (I > I_c_), defined as the herd immunity threshold [23]. The critical prevalence of protected individuals (I_c_) required to establish herd immunity against measles in a completely susceptible population can be determined from: I_c_ = 1 − (1/Ro). This method is based on the following assumptions: (1) a homogeneous mixing of persons within the population, and (2) a homogeneous distribution of protected individuals within the population [23,25].

Measles vaccination programs can reduce measles transmission by means of generating a prevalence of vaccine-induced protected individuals (I_v_) because they reduce the prevalence of susceptible individuals in the population. A lower prevalence of susceptible individuals reduces the basic reproductive number from Ro to the effective basic reproductive number (R) depending on the prevalence of vaccine-induced protected individuals generated by the vaccination program: R = Ro (1 − I_v_). When R is lower than 1, measles transmission is blocked in the community and measles epidemics are prevented. In this situation, imported infections could not reestablish endemic transmission in countries without the circulation of measles viruses. Measles vaccination programs establish herd immunity in the target measles vaccination population when the prevalence of vaccine-induced protected persons is higher than the critical prevalence associated with herd immunity (I_v_ > I_c_) [22,25]. 

The critical measles vaccination coverage associated with herd immunity (V_c_) in the target measles vaccination population was determined from the critical prevalence of protected individuals associated with herd immunity (I_c_) and the effectiveness of the measles vaccination (E): V_c_ = I_c_/E. The critical vaccination coverage associated with herd immunity generates a prevalence of vaccine-induced individuals equal to the critical prevalence of protected individuals (I_v_ = I_c_) for a given measles vaccination effectiveness. The prevalence of vaccine-induced protected individuals in the target measles vaccination population can be determined using the formula: I_v_ = (V_1_× E_1_)+ (V_2_× E_2_). In this formula, V_1_ and V_2_ are the percentages of vaccination coverage with one and two doses of the measles vaccine respectively, and E_1_ and E_2_ are the effectiveness in preventing measles cases with one and two doses of the measles vaccine, respectively. In this study, the critical measles vaccination coverage associated with herd immunity was determined for measles viruses with values of Ro ranging from 6 to 60, and for values of measles vaccination effectiveness ranging from 87.5% to 100%. The range of values considered for measles vaccination effectiveness can cover the values obtained for one and two doses of the measles vaccine [26,27,28]. 

Uzicanin and Zimmerman [26] reviewed the results of studies assessing measles vaccination effectiveness published during 1960−2010, obtaining a 94% vaccination effectiveness for two doses of the measles vaccine (compared with no vaccination), with an interquartile range from 88% to 98%. Marin et al. [27] obtained the values of effectiveness for two doses of the measles vaccine in preventing secondary cases of measles during an outbreak of 95%, with a 95% confidence interval from 82% to 98%. The effectiveness for one dose of the measles vaccine obtained in both studies was 92% [26,27]. The overall routine vaccination coverage and the overall measles vaccination effectiveness depended on the percentage of individuals vaccinated with one and two doses of the measles vaccine. Nevertheless, in Europe, 85.3% of individuals vaccinated during 2017–2019 had received two doses of the measles vaccine, and 86% of measles vaccination effectiveness was generated by two-dose vaccination coverage [9].

Individuals vaccinated with one dose of the measles vaccine who have anti-measles serological levels lower than the protective level (positive anti-measles serological result) are not protected against measles. The lack of measles protection in vaccinated individuals can be explained by primary and secondary vaccination failures. Primary vaccination failure is associated with lack of adequate immune response in vaccinated individuals. Secondary vaccination failure is associated with waning vaccine-induced immunity. Measles vaccination with two doses of the vaccine is associated with a higher effectiveness than vaccination with one dose of the vaccine because it reduces the primary and secondary vaccination failures associated with the first dose of the measles vaccine [19].

The objective of measles vaccination programs can be to achieve a specific effective basic reproductive number (R), or a maximum permissible value of R. The R value must be lower than 1 for achieving measles elimination, but the selection of a specific R value is a policy decision that depends on several factors, including the measles morbidity tolerated, the degree of secondary spread from infected cases tolerated, and the resources available [14]. The critical prevalence of protected individuals associated with a specific effective basic reproductive number, R, can be determined using the formula: I_c_’ = 1 − (R/Ro). The critical vaccination coverage required to achieve a specific R value can be determined using the formula: V_c_ = I_c_’/E. In this study, the critical vaccination coverage (V_c_) required to achieve effective basic reproductive numbers of 0.7 and 0.5 was determined for measles viruses with Ro values ranging from 6 to 60, assuming 95% for two-dose measles vaccination effectiveness.

### 2.2. Critical Measles Seroprevalence Associated with Herd Immunity against Measles Viruses

Anti-measles immunity levels in terms of the prevalence of positive serologic results found in seroprevalence surveys carried out in representative samples of the population were associated with the establishment of herd immunity in the population when they were higher than the critical prevalence of positive measles serologic results (*p* > *p_c_*). In this study, the critical prevalence of positive measles serologic results associated with herd immunity was determined using the following formula: *p_c_* = I_c_ Se + (1 − I_c_) (1 − Sp) [20,25]. In this formula, I_c_ is the critical prevalence of individuals protected against measles associated with herd immunity, and Se and Sp are the sensitivity and specificity of measles serological tests. Values of 97% were assumed for sensitivity and specificity of measles serological tests [20,25]. An alternative formula to estimate the critical prevalence is: *p_c_* = I_c_ Se/PPV [15]. In this formula, PPV is the predictive value of a positive test result. Herd immunity can be considered established when the prevalence of positive measles serologic results obtained in seroprevalence surveys is higher than the critical prevalence associated with herd immunity (*p* > *p_c_*) [15,20,25]. 

### 2.3. Herd Immunity Assessment of the Objectives for Measles Vaccination Coverage and Measles Immunity Levels Proposed by the WHO European Region

Herd immunity levels achieved with the recommended two-dose measles vaccination coverage of ≥95% were assessed by comparing the prevalence of vaccine-induced measles-protected individuals (I_v_) achieved with this vaccination coverage, with the critical prevalence of protected individuals associated with herd immunity (I_c_). The prevalence of protected individuals achieved with 95% two-dose measles vaccination coverage was determined using the formula: I_v_ = V × E. Herd immunity against measles viruses with Ro values ranging from 6 to 60 was considered established when the prevalence of vaccine-induced protected individuals was higher than the critical prevalence (I_v_ > I_c_). 

The European Region of the WHO recommended the following objectives for the proportion of protected individuals (I_who_): ≥85% in children aged 1–4 years, ≥90% in individuals aged 5–9 years, and ≥95% in individuals aged 10–14 years, 15–19 years, and ≥20 years [11,14]. These objectives were designed in order to achieve an effective R = 0.7 against measles viruses, with Ro = 11, assuming a heterogeneous mixing model [14,17]. Based on homogeneous mixing of the population, the critical prevalence of protected individuals required to establish herd immunity against measles viruses with Ro = 11 was 90.9% (9.1% susceptible), and the critical prevalence required to achieve an effective R of 0.7 against measles viruses with Ro = 11 was 93.6% (6.4% susceptible). Therefore, the objective should be 93.6% of protected individuals or 6.4% of susceptible individuals in all age groups, assuming homogeneous mixing. 

The heterogeneous model used by the WHO incorporated variable contact rates among school-aged children [14,17]. The WHO considered that, based on heterogeneous mixing, a ≥95% prevalence of protected individuals (5% susceptible) in individuals aged 10 or more years), allowed lower levels of measles protection or higher levels of susceptibility in children aged 1–4 years (≥ 85%) and children aged 5–9 years (≥90%) [14,17]. In this study, the herd immunity effects derived from achieving the objectives for measles immunity proposed by the WHO were assessed by comparing the proposed prevalence of protected individuals with the critical prevalence of protected individuals-associated herd immunity (R = 1) for measles viruses with Ro values ranging from 6 to 60. Herd immunity was considered established when the prevalence of protected individuals proposed by the WHO was higher than the critical prevalence associated with herd immunity (I_who_ > I_c_). 

The prevalence of positive measles serologic results associated with the measles immunity objectives proposed by the WHO (*p_who_*) were determined using the following formula: *p_who_* = I_who_ Se + (1 − I_who_) (1 − Sp). In this formula, I_who_ is the measles immunity objective, and Se and Sp are the sensitivity and specificity of measles serological tests. Values of 97% were assumed for the sensitivity and specificity of the measles serological tests [20,25]. Herd immunity against measles viruses was considered established when the prevalence of positive measles results associated with the recommended measles immunity levels was higher than the critical prevalence (*p_who_* > *p_c_*). 

## 3. Results

### 3.1. Measles Vaccination Coverage Associated with Herd Immunity against Measles Viruses

Table 2 presents the herd immunity thresholds in terms of the critical prevalence of protected individuals (I_c_), critical vaccination coverage (V_c_), and critical prevalence of positive serologic anti-measles results (p_c_) for measles viruses with basic reproductive numbers (Ro) ranging from 6 to 60. Herd immunity can be considered established when the prevalence of protected individuals is higher than the critical prevalence (I > I_c_), when the vaccination coverage is higher than the critical coverage (V > V_c_), and when the prevalence of positive serologic anti-measles results is higher than the critical prevalence (*p* > *p_c_*). The critical prevalence of protected individuals (I_c_) associated with herd immunity ranged from 83.3% for measles viruses with Ro = 6, to 98.3% for measles viruses with Ro = 60. 

Table 2 shows that the percentages of two-dose measles vaccination coverage from 88% to 100% could establish herd immunity against measles viruses with a Ro ranging from 6 to 19, assuming 95% measles vaccination effectiveness. Measles vaccination programs can establish herd immunity against measles viruses with Ro = 6 by achieving a vaccination coverage higher than the critical coverage of 87.7%, and against measles viruses with Ro = 19 by achieving a vaccination coverage higher than the critical coverage of 99.7%. Nevertheless, measles vaccination programs could not establish herd immunity against measles viruses with Ro ≥ 20, assuming 95% vaccination effectiveness, because the 95% prevalence of vaccine-induced protected individuals associated with 100% vaccination coverage is lower than the critical prevalence of protected individuals required to establish herd immunity.

The critical measles vaccination coverage (V_c_) associated with an effective basic reproductive number, R, of 0.7 ranged from 93% for measles viruses with Ro = 6 to 100% for measles viruses with Ro = 14 (Table 3). Vaccination programs could not establish herd immunity against measles viruses with Ro > 14, assuming 95% vaccination effectiveness, because the prevalence of vaccine-induced protected individuals achieved with 100% coverage (95%) is lower than the prevalence of protected individuals required to establish herd immunity against these viruses. The critical vaccination coverage associated with an effective basic reproductive number, R, of 0.5 ranged from 96.5% for measles viruses with Ro = 6 to 100% for measles viruses with Ro = 10 (Table 3). Vaccination programs could not establish herd immunity against measles viruses with Ro > 10, assuming 95% vaccination effectiveness, because the prevalence of vaccine-induced protected individuals achieved with 100% coverage (95%) is lower than the prevalence of protected individuals required to establish herd immunity against these viruses.

Figure 1 presents the critical vaccination coverage associated with herd immunity against measles viruses with Ro values ranging from 6 to 60, for values of measles vaccination effectiveness ranging from 87.5% to 100%. Figure 1 can be used to estimate the vaccination coverage required to establish herd immunity for different combinations of Ro values for measles viruses and measles vaccination effectiveness. When the effectiveness of the measles vaccination was 90%, herd immunity could be established against measles viruses with Ro < 9, but the vaccination coverage required to establish herd immunity was >95% for measles viruses with Ro values from 7 to 9. When the vaccination effectiveness was 95%, herd immunity could be established against measles viruses with Ro ≤ 20, but the vaccination coverage required to establish herd immunity was >95% for measles viruses with Ro from 10 to 20. 

With a 95% vaccination effectiveness, herd immunity could not be established against measles viruses with high Ro ≥ 21 because in this situation, the prevalence of vaccine-induced measles protection with 100% vaccination coverage is lower than the critical prevalence necessary to establish herd immunity (I_v_ < I_c_). When the vaccination effectiveness was 97%, herd immunity could be established against all measles viruses ranging from 6 to 33, but the vaccination coverage required to establish herd immunity was >95% for measles viruses with Ro from 13 to 33. With a 97% vaccination effectiveness, herd immunity could not be established against measles viruses with high Ro ≥ 34 because in this situation, the prevalence of vaccine-induced measles protection with 100% vaccination coverage is lower than the critical prevalence necessary to establish herd immunity (I_v_ < I_c_).

### 3.2. Measles Seroprevalence Associated with Herd Immunity against Measles Viruses

The critical prevalence of positive serologic results (p_c_) that should be obtained in a serological study when the effective basic reproductive number, R, is 1, ranged from 81.3% for measles viruses with Ro = 6 to 95.4% for measles viruses with Ro = 60 (Table 2). The prevalence of positive serologic results (p_c_) required to establish herd immunity must be higher (at least 0.1% more) than the values obtained for R = 1 because herd immunity can be considered to be established when R is lower than 1. The critical prevalence of positive serologic results that should be obtained when the effective reproductive number is 0.7 ranged from 86% for measles viruses with Ro = 6 to 95.9% for measles viruses with Ro = 60 (Table 3). The critical prevalence of positive serologic results that should be obtained when the effective reproductive number is 0.5 ranged from 89.2% for measles viruses with Ro = 6 to 96.2% for measles viruses with Ro = 60 (Table 3). For measles viruses with Ro = 18, herd immunity can be considered established when the vaccination coverage is 99.5% and the prevalence of positive serologic results is 91.9%.

### 3.3. Herd Immunity Assessment of the Objectives for Measles Vaccination Coverage and Measles Immunity Levels Proposed by the WHO European Region

Figure 1 shows that the objective of 95% for two-dose measles vaccination coverage proposed by the WHO was sufficient to establish herd immunity against measles viruses with Ro values from 6 to 9, but it was not sufficient to establish herd immunity against measles viruses with Ro ≥ 10, assuming a 95% vaccination effectiveness. When the measles vaccination effectiveness increased to 97%, a 95% measles vaccination coverage was sufficient to establish herd immunity against measles viruses with Ro values from 6 to 13. When the measles vaccination effectiveness decreased to 93%, a 95% measles vaccination coverage was sufficient to establish herd immunity only against measles viruses with Ro values from 6 to 8. 

Figure 1 shows that changing the objective for measles vaccination coverage from 95% to 97% could establish herd immunity against measles viruses with Ro values from 6 to 13, assuming 95% measles vaccination effectiveness. When the measles vaccination effectiveness increased to 97%, 97% measles vaccination coverage was shown to be sufficient to establish herd immunity against measles with Ro values from 6 to 18. When the measles vaccination effectiveness decreased to 93%, 97% measles vaccination coverage was sufficient to established herd immunity only against measles virus with Ro values from 6 to 10.

Table 2 shows that the objectives for measles immunity proposed by the WHO in individuals aged 1–4 years (≥85%) and 5–9 years (≥90%) were not sufficient to block the transmission of most measles viruses. By contrast, the objectives proposed in individuals aged 10–14 years, 15–19 years, and ≥20 years (≥95%) were sufficient to block the transmission of measles viruses with Ro values ranging from 6 to 20. In addition, the minimum measles immunity levels proposed in individuals aged 1–4 years (85%) and 5–9 years (90%) were not consistent with the objective for routine two-dose measles vaccination coverage of at least 95%. A 95% vaccination coverage for routine two-dose measles vaccination, given at 12–15 months and 3–15 years, must generate a prevalence of vaccine-induced measles protection of 90.2% in individuals aged 1–4 years and 5–9 years, assuming a measles vaccination effectiveness of 95%. Consequently, I suggest an increase in the objectives for measles immunity in individuals aged 1–4 years and 5–9 years to at least 95% (Table 1).

## 4. Discussion

The main objectives of measles vaccination programs developed by public health departments are to protect vaccinated individuals against measles and to achieve sufficient vaccination coverage to establish the necessary herd immunity to prevent measles transmission in the population. This study found that success in preventing measles transmission and achieving measles elimination depended on the basic reproductive number (Ro) of the measles viruses and measles vaccination effectiveness. The WHO proposed to achieve and maintain percentages of measles vaccination coverage with two doses of measles vaccines equal to or higher than 95%, as well as to achieve specific anti-measles immunity levels in different age groups. Nevertheless, this study found that the objectives proposed by the WHO for two-dose measles vaccination coverage and measles immunity were not sufficient to achieve measles elimination in Europe.

The study determined the minimum vaccination coverage required to establish herd immunity against measles viruses with Ro values ranging from 6 to 60. High percentages of two-dose measles vaccination coverage are necessary to establish herd immunity and block measles transmission [20,22,25]. The study found that the minimum two-dose measles vaccination coverage recommended by the WHO (95%) was not sufficient to establish herd immunity against measles viruses with Ro values equal to or higher than 10, assuming a 95% vaccination effectiveness. In addition, the recommended two-dose vaccination coverage was not sufficient to establish the herd immunity required to achieve an R value of 0.7 against measles viruses with Ro values equal to or higher than 8, and it was not sufficient to establish the herd immunity required to achieve an R value of 0.5 against measles virus with Ro values equal to or higher than 6. Consequently, I suggest an increase in the recommended two-dose vaccination coverage to ≥97% (Table 1).

The study determined the minimum proportion or prevalence of protected individuals required to establish herd immunity against measles viruses with Ro values ranging from 6 to 60. High percentages of measles immunity are necessary to establish herd immunity and block measles transmission [17,20,29]. The study found that the minimum proportion or prevalence of protected individuals recommended by the WHO in children aged 1–4 (85%) and individuals aged 5–9 years (90%) were not sufficient to establish herd immunity against most measles viruses. By contrast, the prevalence of protected individuals recommended in individuals aged ≥10 years (95%) was sufficient to establish herd immunity against measles viruses. Consequently, I suggest an increase in the recommended prevalence of measles protection to at least to 95% in individuals aged 1–4 and 5–9 years (Table 1).

The recommendations proposed in this study for measles immunity in different age groups were determined for measles viruses with Ro values from 6 to 60 assuming a homogeneous mixing of the population. The objectives proposed by the WHO in children aged 1–4 and 5–9 years were designed for measles viruses with an Ro value equal to 11 using a heterogeneous mixing model. These criteria allowed a lower prevalence of measles protection in children aged 1–4 years (≥85%) and individuals aged 5–9 years (≥90%) [17,30]. 

The objectives for measles immunity should be increased to at least 95% in individuals aged 1–4 and 5–9 years for two reasons. First, the WHO’s recommendations for measles immunity were determined assuming a Ro value of 11 for measles viruses, but measles viruses can be associated with values of Ro higher than 11. Second, the minimum measles immunity levels proposed in individuals aged 1–4 years (85%) and 5–9 years (90%) were lower than the prevalence of vaccine-induced measles protection of 90.2% generated by 95% two-dose measles vaccination coverage, assuming a measles vaccination effectiveness of 95%.

This study has several limitations. First, the herd immunity assessment of the objectives for two-dose measles vaccination coverage and measles immunity levels proposed by the WHO was carried out based on the following assumptions: (1) homogeneous mixing of individuals within the population, and (2) homogeneous distribution of protected individuals within the population. The approach used in this study is based on the following reasons: (1) the mixing pattern and contact rates among individuals is not known, and (2) assuming a heterogeneous mixing can reduce the recommended two-dose vaccination coverage and measles immunity in population groups without a consistent justification. The heterogeneous mixing model reduces the vaccination coverage required to establish herd immunity in age groups with low measles transmission rates (1–9 years), when compared to the coverage based on homogeneous mixing. Nevertheless, individuals aged 1–9 years are the target population for routine measles vaccination and have the highest risk for measles complications [4,9,21]. For this reason, the herd immunity assessment based on the homogeneous mixing model can be considered a conservative approach.

The second limitation is that the herd immunity assessment for different percentages of two-dose measles vaccination coverage was carried out assuming vaccination effectiveness ranging from 87.5% to 100%. The establishment of herd immunity would be more difficult for values of measles vaccination effectiveness lower than 87.5%; however, it is possible to assume that 87.5% is the lower effectiveness for two-dose measles vaccination programs [26,27,28].

The results obtained in this study indicated that it is necessary to increase the recommended objectives for measles vaccination coverage with two-dose coverage in order to achieve measles elimination from Europe from ≥95% to ≥97%. The current objectives for measles vaccination coverage recommended by the WHO for routine measles vaccination could be sufficient for achieving measles elimination, however, with intensive supplementary measles vaccination activities based on mass catch-up and follow-up vaccination campaigns. Supplementary mass catch-up vaccinations should be given to children, adolescents, and adults who have not received the first or second dose of measles vaccine or who have lost their vaccination records [4]. Catch-up vaccination activities are usually implemented for a limited time. The measles prevention strategy based on high routine measles vaccination of children and intensive supplementary vaccination activities was implemented by the Pan American Health Organization (PAHO) in 1980–1990, and it succeeded in interrupting the transmission of measles in America [4,31]. The objective of the PAHO was to achieve at least 95% coverage with the MMR vaccine for children aged 12 months and to implement intensive supplementary vaccination activities to ensure the establishment of herd immunity in all countries. 

The intensive supplementary vaccination activities consisted of one mass campaign among adolescents and adults, catch-up campaigns in children aged 1–14 years, and follow-up campaigns in children aged < 5 years. In Europe, the objective for measles vaccination coverage with two-dose coverage should be increased from ≥95% to ≥97% as intensive supplementary vaccination activities or screening and vaccination programs are not developed. In addition, measles prevention based on routine and supplementary vaccination activities requires continuous intensive catch-up vaccination activities to maintain high percentages of vaccination coverage [32].

An alternative strategy to increase measles immunity in the population can be based on the administration of a third dose of the measles vaccine to young adults (15–35 years). Two vaccination programs could be developed: (1) vaccination of susceptible young adults detected by pre-vaccination screening (catch-up of susceptible), and (2) vaccination of all young adults, regardless of their vaccination and immunity status (catch-up or routine vaccination) [20]. Nevertheless, increasing routine two-dose measles vaccination to ≥97% has a higher priority than vaccinating young adults with a third dose of measles vaccine for two reasons. First, vaccinating young adults with a third dose of measles vaccine will not increase measles protection in the target vaccination population (1–9 years). Second, the strategy based on a third dose of measles vaccine in young adults is more complex and costly than increasing two-dose measles vaccination coverage to ≥97%. 

Routine measles vaccination coverage with two doses of the vaccine could be increased to 97% by implementing interventions to enhance access to vaccination services, interventions to increase demand for vaccines, interventions to increase provider vaccination activities, and interventions to increase vaccine confidence [33,34,35]. Specific regulations on vaccination requirements can increase measles vaccination coverage, but this strategy should add to, more than replace, other strategies to reach and maintain high percentages of measles vaccination coverage [36]. In the European Union, measles vaccination (the MMR vaccine) was mandatory in 28% of the countries in 2010 [37]. Advanced vaccination programs, based on nominal vaccination registries and health information technology, can implement interventions to increase measles vaccination coverage in a more efficient way than less advanced vaccination programs [38,39].

## 5. Conclusions

This study found that the objectives for two-dose measles vaccination coverage and measles immunity levels in children recommend by the WHO European Region were not sufficient to establish herd immunity against measles viruses with a Ro equal to or higher than 10. To meet the goal of measles elimination in Europe, it is necessary to achieve percentages of two-dose measles vaccination coverage of at least 97%, and measles immunity levels in children aged 1–9 years of at least 95%.

## Figures and Tables

**Figure 1 vaccines-08-00218-f001:**
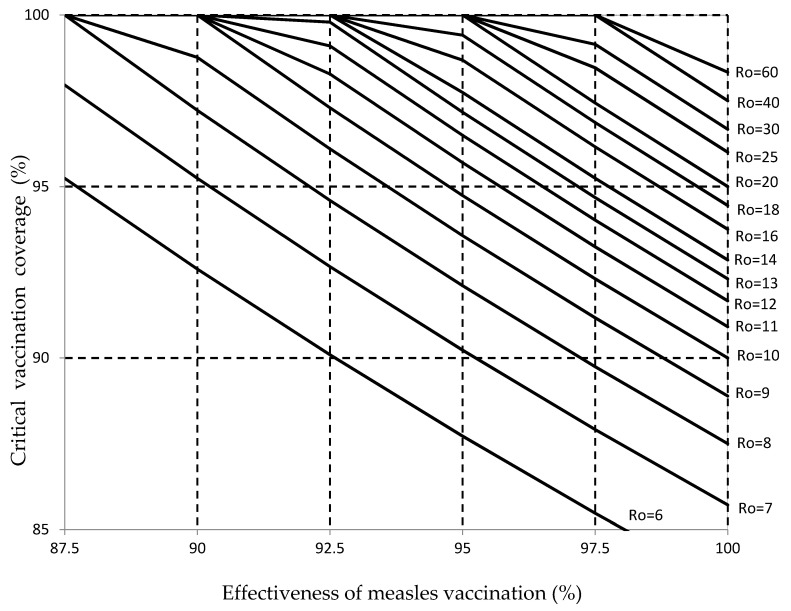
The vaccination coverage (%) required to establish herd immunity for measles viruses with reproductive numbers (Ro) from 6 to 60 and the effectiveness of measles vaccination from 87.7% to 100%. The objectives of measles vaccination coverage with two doses of vaccine of 95% (proposed by the WHO European Region at the national level), and 90% are indicated by horizontal dashed lines.

**Table 1 vaccines-08-00218-t001:** Objectives for two-dose measles vaccination coverage during childhood and for the proportion of protected individuals in different age groups proposed by the World Health Organization (WHO) and the objectives proposed in this study.

Objectives for Measles Vaccination Coverage and Measles Immunity	Proposed by the WHO [4,13,14]	Proposed in This Study
Two-dose measles vaccination coverage during childhood	≥95%	≥97%
Proportion of individuals protected against measles (% susceptible)		
Aged 1–4 years	≥85% (≥15%)	≥95% (≥5%)
Aged 5–9 years	≥90% (≥10%)	≥95% (≥5%)
Aged 10–14 years	≥95% (≥5%)	≥95% (≥5%)
Aged 15–19 years	≥95% (≥5%)	≥95% (≥5%)
Aged ≥ 20 years	≥95% (≥5%)	≥95% (≥5%)

**Table 2 vaccines-08-00218-t002:** Herd immunity thresholds in terms of the critical prevalence of protected individuals (I_c_), critical vaccination coverage (V_c_), the percentage of susceptible individuals in vaccinated populations, and the critical prevalence of positive anti-measles serologic results (*p_c_*) for measles viruses with basic reproductive numbers (Ro) ranging from 6 to 60.

Measles VirusRo	Critical Prevalence of Protected Individuals ^a^ I_c_ (%)	Measles Vaccination		Critical Prevalence of Positive Serologic Results ^d^ *p_c_* (%)
		Critical Vaccination Coverage ^b^ V_c_ (%)	Susceptible Individuals ^c^ (%)	
6	83.3	87.7	16.7	81.3
7	85.7	90.2	14.3	83.6
8	87.5	92.1	12.5	85.3
9	88.9	93.6	11.1	86.6
10	90.0	94.7	10.0	87.6
11	90.9	95.7	9.1	88.5
12	91.7	96.5	8.3	89.2
13	92.3	97.2	7.7	89.8
14	92.9	97.7	7.1	90.3
15	93.3	98.2	6.7	90.7
16	93.8	98.7	6.2	91.1
17	94.1	99.1	5.9	91.5
18	94.4	99.4	5.6	91.8
19	94.7	99.7	5.3	92.1
20	95.0	100	5	92.3
21	95.2	−	5	92.5
22	95.5	−	5	92.7
23	95.7	−	5	92.9
24	95.8	−	5	93.1
25	96.0	−	5	93.2
30	96.7	−	5	93.9
40	97.5	−	5	94.7
50	98.0	−	5	95.1
60	98.3	−	5	95.4

^a^ I_c_ = 1 − (1/Ro), ^b^ V_c_ = I_c_/E. Assuming 95% effectiveness (E) of 95% for measles vaccination. ^c^ Percentage of susceptible individuals = 100 − (V_c_ E), ^d^
*p_c_* = I_c_ Se + [(1 − I_c_) (1 − Sp)]. Assuming 97% sensitivity (Se) and 97% specificity (Sp) for measles serological tests.

**Table 3 vaccines-08-00218-t003:** The critical prevalence of protected individuals (I_c_), critical vaccination coverage (V_c_), and critical prevalence of positive anti-measles serologic results (p_c_) associated with an effective basic reproductive number, R, of 0.7 and 0.5, for measles viruses with basic reproductive numbers (Ro) ranging from 6 to 60.

Measles Virus Ro	Herd Immunity Thresholds I_c_, V_c_, and *p_c_* Associated with an Effective Basic Reproductive Number, R ^a^, of 0.7 and 0.5					
	R = 0.7			R = 0.5		
	I_c_ (%) ^b^	V_c_ (%) ^c^	*p_c_ (%)* ^d^	I_c_ (%) ^b^	V_c_ (%) ^c^	*p_c_* (%) ^d^
6	88.3	93.0	86.0	91.7	96.5	89.2
7	90.0	94.7	87.6	92.9	97.7	90.3
8	91.3	96.1	88.8	93.8	98.7	91.1
9	92.2	97.1	89.7	94.4	99.4	91.8
10	93.0	97.9	90.4	95.0	100	92.3
11	93.6	98.6	91.0	95.5	−	92.7
12	94.2	99.1	91.5	95.8	−	93.1
13	94.6	99.6	91.9	96.2	−	93.4
14	95.0	100	92.3	96.4	−	93.6
15	95.3	−	92.6	96.7	−	93.9
16	95.6	−	92.9	96.9	−	94.1
17	95.9	−	93.1	97.1	−	94.2
18	96.1	−	93.3	97.2	−	94.4
19	96.3	−	93.5	97.4	−	94.5
20	96.5	−	93.7	97.5	−	94.7
21	96.7	−	93.9	97.6	−	94.8
22	96.8	−	94.0	97.7	−	94.9
23	97.0	−	94.1	97.8	−	95.0
24	97.1	−	94.3	97.9	−	95.0
25	97.2	−	94.4	98.0	−	95.1
30	97.7	−	94.8	98.3	−	95.4
40	98.3	−	95.4	98.8	−	95.8
50	98.6	−	95.7	99.0	−	96.1
60	98.8	−	95.9	99.2	−	96.2
Mean	95.3	99.0	92.5	96.6	99.7	93.8

^a^ The effective basic reproductive number (R) is the reduced Ro number due to vaccine-induced protected individuals (I_v_). R = Ro (1 − I_v_), ^b^ I_c_ = 1 − (R/Ro), ^c^ V_c_ = I_c_/E. Assuming 95% effectiveness (E) for measles vaccination, ^d^
*p_c_* = I_c_ Se + [(1 − I_c_) (1 − Sp)]. Assuming 97% sensitivity (Se) and specificity (Sp).

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
