# Peer review of "Are the Objectives Proposed by the WHO for Routine Measles Vaccination Coverage and Population Measles Immunity Sufficient to Achieve Measles Elimination from Europe?"

_vaccines, 2020, doi:10.3390/vaccines8020218_

Round 1
Reviewer 1 Report
The article is a good theoretical exercise on the rate of vaccination coverage necessary to guarantee herd immunity against measles. However, there are serious concerns that the model applies to reality, due to the limitations that the author himself recognizes, and others that will be mentioned ahead. Although the author is careful to mention that his conclusions refer to the age group of 1-9 years old, measles does not occur only at these ages.
The recent measles cases that occurred in Europe do not only appears in childhood (1-9 years), but also at older ages, so is necessary to understand:
If individuals vaccinated longer have the same probability of protection against disease as those vaccinated recently;
If individuals with serological levels below the one considered protective, and vaccinated with 2 doses, are equally protected against measles;
Whether it is more effective to administer a 3rd dose of measles vaccine to young adults or to increase the vaccination coverage rate by 2% as suggested by the author.
It is considered essential to clarify these aspects taking into account the epidemiology of measles in Europe and when the aim is to eliminate the disease.
Author Response
Thank you for your comment.
Point 1. The article is a good theoretical exercise on the rate of vaccination coverage necessary to guarantee herd immunity against measles. However there are serious concerns that the mkodel applies to reality, due to the limitations that the same author himself recognizes, and other that will be mentioned ahed. Although the autor is careful to mention this conslusions refres to the age group of 1-9 years old, measles does not occur only at these ages.
Response to point 1: The heterogeneous mixing model reduces the vaccination coverage required to establis herd immunity ina ge groups with low measles transmisison rates (1-9 years), when compared to the vaccination coverage based on homogeneous mixing. Nevertheless, individuals aged 1-9 years nare the target population for routine measles vaccination, and have the highest risk for measles complications. For this reason, the herdc immunity assessment based on homogeneous mixing model can be considered a conservative approach.
I have included this explanation on lines 420-426 of the revised paper.
Point 2. It si neccesary to understand: If individuals with serological levels bellow the one considered protective,a nd vaccinated with 2 doses, are equallt protected against measles.
Response to point 2: Individuals vaccinated with one dose of measles vaccine who have anti-measles seroplogical levels lower than the level considered as protective (positive anti-measles serologic result) are not protected against measles. The lack of mesasles protection in vaccinated individuals can be explained by primary and secondary vaccination failures. Primary vaccination failure is associated with lack of adequate immune response in vaccinated individuals. Secondary vaccination failure is associated with waning vaccine-induced immunity. Measles vaccination with two doses of vaccine is associated with a higher effectiveness than vaccination with one dose because it reduces the primary and secobdary vaccination failures assciated with the first dose of measles vaccine.
I have include this explanation on lines 180-187 of the revised paper.
Point 3. Whether it is more effective to administe a 3rd dose of measles to young adults or to increase the vaccination coverage rate by 2% as suggested by the author.
Response to point 3: An alternative strategy to increase measles immunity in the population can be based on the administration of a third dose of measles vaccine to young adults (15-35 years). Nevertheless, increasing routine two-dose measles vaccination to at least 97% has a higher priority than vaccinating young adults with a third dose of measles vaccine for two reasons: First, vaccinating young adulst with a third dose of measles vaccine will not increase measles protection in the target vaccination population (1-9 years). Second, the strategy based on a third dose of measles vaccine in young adults is more complex and costly than increasing two-dose measles vaccination coverage to 97% or more.
I have included this explanation on lines 453-463 of teh revised paper.Â
Â
Â
Â
Â
 Â
Reviewer 2 Report
I was invited to review the paper entitled "Are the objectives proposed by the WHO for routine 2 measles vaccination coverage and population measles 3 immunity sufficient to achieve measles elimination 4 from Europe?". THe aim of this study was assessing whether the objectives proposed by the WHO for measles vaccination coverage and antimeasles immunity are sufficient to establish herd immunity. It is a very interesting topic for the field It deeply discuss the herd immunity assessment of the objectives for measles vaccination coverage and measles immunity levels. Methods of assessment were well described and results were reported clearly. Despite these points, I have to raise some minor points:
- The Authors should discuss about the history of measles vaccine. In particular, in Europe this vaccine was introduced during last 80s years, with a single dose strategy. This is one of the main cause of the high incidence of measles among adults during last years in Europe;
- The Author discussed about measles outbreaks happened in Europe during last years. I suggest to discuss all possible causes of them and strategies of European countries performed to improve VC;
- Line 376: the first sentence was reported twice.
Author Response
Thank for your comments.
Point one. The authors should discuss about the hsitory of measles vaccine.
Response to point 1. I have include on lines 40-46 of the revised paper a discussion on the history of measles vaccine.
In the WHO European Region, the mean measles vaccination coverage for the fist dose of measles vaccine increased from 76.8% in 1980 to 92-94% since 2003; and the mean vaccination coverage for the second dose of measles vaccine increased from 68% in 1980 to 91-92% since 2004. The mean vaccination coverage with two doses of measles vaccine increased from 58.6% in 1995 to 84-88% since 2004.
Point 2. I suggest to discuss all possible causes of measles persistence in Europe and strategies of European countries performed to improve vaccination coverage.
Response to point 2: I have included on lines 63-68 of the revised paper a discussion on the factors expalineg measles persistance in Europe and the stragies developed by European countries to increase vaccination coverage.
The factors explaining persistence of measles in Europe include: low measles vaccination coverage with two doses of vaccine; low anti.measles immunity levels in areas and population groups (immunity gaps); mobility of individuals with measles across Europe; and loss of public confidence in vaccines. Countries of the eastern part of Europe conducetd supplementary vaccination activities, but they were not enough to achieve the vaccination coverage required to block measles transmisison in Europe. Â
Point 3. Line 376: the first sentence was reported twice.
Response to point 3: The sentence was corrected.
Â
Reviewer 3 Report
The review presents available data on the circulation of the measles virus and the effectiveness of the vaccination levels necessary to achieve the goal of elimination. The suggestion that 97% vaccination coverage should be achieved is acceptable.
Author Response
Thank you for your comments.